# The Impact of COVID-19 on Telemedicine Utilization Across Multiple Service Lines in the United States

**DOI:** 10.3390/healthcare8040380

**Published:** 2020-10-01

**Authors:** Jose A. Betancourt, Matthew A. Rosenberg, Ashley Zevallos, Jon R. Brown, Michael Mileski

**Affiliations:** School of Health Administration, Texas State University, San Marcos, TX 78666, USA; mar401@txstate.edu (M.A.R.); anz11@txstate.edu (A.Z.); jrb296@txstate.edu (J.R.B.); mileski@txstate.edu (M.M.)

**Keywords:** telemedicine, COVID-19, telehealth, health service lines, pandemic

## Abstract

The impact of COVID-19 on the U.S. healthcare industry cannot be overstated. Telemedicine utilization increased overnight as all healthcare providers rushed to implement this delivery model to ensure accessibility and continuity of patient care. Our research objective was to determine measures that were implemented to accommodate community and individual patient needs to afford access to critical services and to maintain safety standards. We analyzed literature since 2016 from two databases using Preferred Reporting Items for Systematic Reviews and Meta-Analyses (PRISMA). We compared observations, themes, service lines addressed, issues identified, and interventions requiring in-person care. From 44 articles published, we identified ten effectiveness themes overall and drew conclusions on service line successes. COVID-19 has caused rapid expansion in telemedicine. Necessary and required changes in access, risk mitigation, the need for social distancing, compliance, cost, and patient satisfaction are a few of the driving factors. This review showcased the healthcare industry’s ability to rapidly acclimate and change despite the pervasive spread of COVID-19 throughout the U.S. Although imperfect, unique responses were developed within telemedicine platforms to mitigate disruptions broadly and effectively in care and treatment modalities.

## 1. Introduction

The use of telemedicine is neither a recent phenomenon nor only found in the United States. One need not look any further than the islands of southern Italy, where a successful telemedicine program has provided great improvements to healthcare delivery to remote areas, bringing with it high-quality care. This telemedicine solution has provided a prompt and qualified health service in the islands while reducing the risks and costs of patient transportation to the mainland [1]. In the United States, an April 1924, Radio News magazine foreshadowed the idea of modern telemedicine through the depiction of a “radio doctor” linked to a patient via sound and a live picture [2]. In 1948, the first known transfer of records through telephone lines occurred in Pennsylvania, when radiologic images were sent across a 24-mile stretch between West Chester and Philadelphia [3]. A decade later, in 1959, medical uses of video communications were initiated at the University of Nebraska, when they used two-way interactive television to transmit information across campus and, in 1964, further established a telemedicine link with the Norfolk State Hospital 112 miles away for group consultations through a closed-circuit, two-television link [2]. A breakthrough in urban emergency medicine was initiated in 1967, between the University of Miami School of Medicine and local fire departments to transmit electrocardiographic rhythms in rescue situations over the radio to Jackson Memorial Hospital [4]. In 1968, a foundational telemedicine project was established involving Boston’s Logan Airport, and nearby Massachusetts General Hospital (MGH), in which a medical station at the airport was linked via microwave relay to the hospital as a vehicle for providing primary and emergency services to airport staff and travelers. MGH parlayed the success of this effort two years later with a tele-psychiatry link to the Veteran’s Administration Hospital in Bedford, MA, that continued operations into the 1980s [5]. One of the earliest joint ventures involving a federal agency was the Space Technology Applied to Rural Papago Advanced Health Care Project (STARPAHC), which was a major initiative begun in 1973, between what was then known as the U.S. Department of Health, Education, and Welfare (now DHHS—Department of Health and Human Services), and involved National Aeronautics and Space Administration (NASA), Lockheed Company, and the U.S. Indian Health Service, in which they began sharing complex medical data, including neurological examinations, videos, stethoscope sounds, and radiological images, to promulgate healthcare services to individuals in distant locales [6].

With the advent of the Internet in the 1990s, telemedicine capabilities proliferated as networks began integrating on a global scale and rendered an easier and cheaper method of building software applications specifically geared towards the asynchronous exchange and storage of clinically-relevant data through mediums such as e-mails, text messaging, and patient portals, and the subsequent synchronous interfacing between provider and patient in real time with live video feeds. Electronic medical records, with the assistance of government incentives, are now ubiquitous and can be engineered to include clinical metrics and user-friendly interfaces to optimize patient care, telemedicine capabilities, and interoperability with vendors, such as laboratory and imaging organizations, for workflow proficiency.

### 1.1. Definition of Key Terms

The Kaiser Family Foundation (2020) considers telehealth and telemedicine to be interchangeable terms and collectively defines them as the “remote provision of healthcare services using technology to exchange information for the diagnosis, treatment, and prevention of disease.” [7] Ludwig and Zarbock (2020) delineate Severe Acute Respiratory Syndrome-Associated Coronavirus 2 (SARS-CoV-2) as the novel strain of coronavirus and it is defined as the causal agent of Coronavirus Disease 2019 (COVID-19), these terms may also be interchangeable when referencing the current, ongoing pandemic [8]. DHHS is an acronym for the Department of Health and Human Services, which is the executive branch department of the U.S. Federal Government responsible for providing essential human services to advance the health and well-being of the American people. CMS stands for the Center for Medicare and Medicaid Services, which is an agency embedded within the DHHS that administers the major health programs in the U.S.

### 1.2. Rationale

Table 1 illustrates the transformation in both patient and provider utility of telemedicine both prior to and following the COVID-19 global pandemic. In many instances, the pandemic served as a catalyst accelerating the overall “acceptance” as a viable means of healthcare delivery in the eyes of both patient and provider. 

Accommodations in federal, state, and local jurisdictions have facilitated the advancement of telemedicine capabilities through the relaxation of state licensure requirements, DHHS enforcement discretion and penalty waivers of HIPAA (Health Insurance Portability and Accountability Act of 1996)regulations, dissemination of reimbursement allowances for current procedural terminology (CPT) codes, modifiers, and appropriate clinical encounter documentation, and general acceptance and provision of telemedicine utilization to enable the preservation of clinical service delivery [13].

### 1.3. Significance

The swift rise of COVID-19 infection rates upended traditional care delivery models, which overwhelmingly centered around in-person visitations within inpatient and ambulatory care settings. Restrictive measures enacted by state and local officials, such as stay-at-home orders, necessitated the expansion of a nascent platform to accommodate routine and emergent needs for patient populations. Telemedicine can never fully replace face-to-face clinical encounters due to its unconventional and remote nature, particularly as it relates to clinician-assisted care provisions such as physical exams, vaccinations, lab work, imaging, rehabilitative therapies, etc. Despite the absence of hands-on engagement, telemedicine’s widespread acceptance and utilization in the wake of the COVID-19 pandemic highlights it as a meaningful and valuable model of care for current and future practice [14].

### 1.4. Objective

Telemedicine utilization erupted on an unprecedented scale as healthcare providers across the U.S. endeavored to stem the deleterious effects of COVID-19. Although the healthcare industry in general was significantly impacted on various levels, we strived to answer this specific question: “What impact has the current COVID-19 pandemic had on the provision of care through telemedicine across unique health service lines that include dermatology, oncology, obstetrics/gynecology, and mental health?”

## 2. Methods 

### 2.1. Protocol and Registration

This review used the Preferred Reporting Items for Systematic Reviews and Meta-Analyses (PRISMA) format. The review was not registered, as the registration was not completed prior to analysis per PROSPERO rules. PROSPERO is an international database of prospectively registered systematic reviews in health and social care.

### 2.2. Eligibility Criteria

Studies were deemed germane for this review if articles mentioned the effects of COVID-19 and telemedicine utilization to specific service lines in healthcare to include dermatology, oncology, obstetrics/gynecology, and mental health. These specific service lines were chosen due to the unique nature of their treatment modalities, being often not associated with telemedicine. Mental health was included as telemedicine is used as a normal and expected treatment modality and would provide for excellent comparison to the others chosen. Researchers examined these service lines further due to their differing approaches to patient care and the adaptations that were necessary to obtain desired outcomes from a patient and provider perspective. Articles included were published between 2016 and 2020.

### 2.3. Information Sources

Two databases were queried: The Cumulative Index of Nursing and Allied Health Literature (CINAHL) and PubMed (MEDLINE). Databases were filtered for articles between 2016 and 2020. Database searches occurred during June 2020.

### 2.4. Search

Reviewers carefully analyzed results from the search across two databases using specific tailored Boolean operators in each. Identical Boolean searches were used in each database. The combination of terms yielded the maximum number of results in both databases. Researchers each reviewed abstracts of articles to determine their relevance to the review and to identify overarching themes. 

### 2.5. Study Selection

Search results from two databases were downloaded to a common Excel spreadsheet which was used as a literature matrix. This spreadsheet continued to be used throughout the process to extract data and analyze results. Researchers met weekly to refine search terms, discuss findings, assign workload, and troubleshoot barriers. Within several of these collaborative sessions, consensus meetings were held to establish final agreement on articles germane to our research that would comprise the foundation of our systematic literature review. Author agreement was ultimately measured via calculation of a kappa statistic which was 0.78, showing substantial agreement among authors.

PRISMA guidelines were used to keep track of the processes of narrowing the initial search results of 52,206 articles to the final selection of 44 sources. Filters were applied to restrict results to a publication date between 2016 and 2020 and the English language. After filtering and removing duplicates, the authors reviewed the abstracts of 52,164 articles. Only those germane to this review, which discussed relationships between COVID-19 and telemedicine surrounding specific service lines, were included in the final selection. All authors collaborated to reach a consensus on the inclusion of articles in the final selection of 44.

### 2.6. Data Collection Process

Each germane article was reviewed by at least two reviewers who agreed that each article included was germane to the study. Reviewers read through articles twice to ensure that articles were germane and that we were able to make observations relative to the objective of the study. From these observations, a thematic analysis was performed to better organize and make sense of the data. These themes were then used to draw inferences from each of the germane articles and to draw conclusions regarding specific service lines.

### 2.7. Data Items

Via use of the spreadsheet, standard data items were collected: observations, themes, service lines addressed, issues identified in the articles surrounding the use of telemedicine, and any information discussing things requiring in-person care.

### 2.8. Risk of Bias in Individual Studies

SARS-CoV-2 is an evolving global concern and data continues to be collected for literature that is yet to be written and, subsequently, circulated. Resulting from this limited resource availability, what we identified in this review included editorials and other published articles based on best practice recommendations not necessarily applicable to all health service line domains. This may lead to an inherent bias within an article’s thesis if it is argued through a lens that is not tested in a practical or real world setting and only speaks to a narrowly defined specialty or primary care arena.

The research group incorporated strategies to minimize the effects of the articles’ inherent bias. This included the assignment of article reviews on a randomized basis to which each member had a nearly identical volume of articles to evaluate. Independent analysis was conducted and insights were subsequently shared through a data repository accessible to each member.

### 2.9. Summary Measures

The review analyzed studies with qualitative, quantitative, mixed methods, editorials, and other published sources, as such the summary measures sought were not consistent. A preferred summary statistic for this study would have been a risk ratio, however, due to the nature of the literature reviewed, this was not possible to measure.

### 2.10. Additional Analysis

During author consensus meetings, a thematic analysis was performed to group observations from the literature into themes. These themes were measured across all literature analyzed and reported as summary statistics in the affinity matrix. The thematic analysis identified ten themes, as reported in Table 2.

## 3. Results

### 3.1. Results of Individual Studies and Synthesis of Results

The impact of the COVID-19 pandemic has created a rapid expansion in telemedicine administration. To reduce the spread of the virus, healthcare providers have engaged social distancing and heightened infection control measures with their patients [15,16,17,18,19,20,21,22,23,24,25,26,27,28]. Telemedicine has been effective through risk mitigation, improved access, convenience, lower cost, and patient satisfaction, of which improved access and risk mitigation were the leading themes [1,13,15,16,17,20,21,22,23,24,25,27,28,29,30,31,32,33,34,35,36,37,38,39,40,41,42,43,44]. The results have also shown a relaxation of licensure requirements, thereby allowing providers to practice across state lines, as well as with HIPAA regulations so devices including smartphones and tablets can be utilized for video conferencing within applications such as Zoom, FaceTime, and Google Hangouts Meet [13,21,27,32,34,39,41,45,46]. From the four selected service lines of dermatology, mental health, OB/GYN (obstetrics and gynecology), and oncology, mental health has had the most documented outcomes with the use of telemedicine. This is due in large part because mental health is conversational in nature and the provision of care does not hinge upon in-person interactions. Each service line has found effective uses of telemedicine during the COVID-19 pandemic and, although there are still areas that will require in-person visits for testing, ultrasounds, physical exams, etc., telemedicine is demonstrating that many visits do not require physical attendance and has significantly reduced the total number of patients being seen at a healthcare facility.

Figure 1 illustrates the PRISMA roadmap that comprised our search and selection process. In June 2020, the researchers created a collection of search strings to gather the necessary amount of quality source material and narrow the scope of the search, which led to a sizable return of results. The initial search yielded 267 articles in CINAHL and 51,939 in PubMed, for a cumulative count of 52,206. After applying our limiters, we pared down the final tally to 15 for CINAHL and 29 for PubMed.

Table 2 summarizes the main observations in the articles used and common themes expressed in the investigation that pertained to the search query. The “Outcomes” section did not apply to most of the articles since they are mainly based on opinion pieces; however, articles 15 and 13 were based on studies and the results can be seen in the column. Articles 1–2, 4–5, 8–10, and 16 were used for historical evidence and did not impact the results. 

### 3.2. Additional Analysis 

Table 3 is an affinity matrix that displays the effectiveness themes extracted from our research and the specific articles in which they were found. The total number of occurrences were summed and divided by the total volume of articles (*n*) to create the probability of occurrence. It should be noted these themes are meant to demonstrate a measure of mutual similarity and do not equate to a hierarchy of importance.

Table 4 is a chart that outlines some of the implementation successes of telemedicine utilization on the health service lines we selected for this systematic literature review in the wake of COVID-19 proliferation. Each service line experiences diverse workflow requirements and care provision criteria, which translated to varying levels of positive outcomes resulting from the accelerated transition to virtual medicine.

## 4. Discussion

### 4.1. Summary of Evidence

Telemedicine acts as a bridge between patient and provider in separate locations in one of the safest manners possible. The sweeping prevalence of COVID-19 within the U.S. forced the healthcare industry to reimagine their models of care and how they provisioned services to new and established patients that directly resulted in the explosion of telemedicine utilization. We narrowed our focus to address four specific service lines with unique requirements to examine how disruptions to continuity of care could be mitigated, which required sophisticated, timely course corrections to meet individual and community needs for urgent and non-urgent encounters.

A detailed analysis of our research resulted in the emergence of an array of themes that were commonplace within the selected service lines. The leading indicators for telemedicine utilization expansion were efforts to increase access to patients, deploy social distancing measures, and attempt to mitigate the risks of exposure to other patients. By virtue of the telemedicine platform, patients were less encumbered by geographic distance, particularly those residing in rural areas, as well as physical limitations, and transportation concerns, thereby allowing them to engage their providers outside of the clinical space. In effect, this increased access enabled a dual track of social distancing allowance and risk mitigation that precluded concerns of having to avoid contact with other patrons in lobbies, hallways, exam rooms, restrooms, etc., where an exchange of microbes is commonplace.

Telemedicine provides a more convenient method of care provision as patients can obtain services within the confines of their preferred location whether its their home, office, car, etc. This reduces the burdens of travel time, missed work and/or school, and opportunity and monestary costs.

Two of the more notable barriers to widespread telemedicine adoption prior to COVID-19 dispersion were issues related to HIPAA compliance in the context of privacy concerns with secure communication lines and data sharing, burdensome regulations, interstate licensure requirements, and reimbursement disparities amongst U.S. insurers who rendered payouts not on-par with in-person visits. These obstacles were relaxed by CMS, DHHS, and public and commercial payers to accommodate a new care provision reality. Even though telemedicine was marginally utilized prior to the spread of COVID-19, additional training and education for patients and providers are necessary to adapt to changing protocols, assessment criteria, and basic understanding of telemedicine functionality [13,17,19,20,35,37,38,42,45,46]. Widespread infection rates and spikes in hospitalizations resulting from COVID-19 proliferation spurred the rapid adoption of telemedicine utilization virtually overnight. While this was aided and abetted by local stay-at-home orders and the broader public fears of acquiring the disease, healthcare providers realized the need to still provide critical services to their patient populations and understood telemedicine platforms were the perfect vehicles to rise to the occasion. A direct result of this expansion was the need to fully educate providers, clinical/administrative support staff, and patients in how to effectively utilize this new model of care provision. This included webinars and online tutorials, quick reference guides, phone consultations, and do-it-yourself learning. Lastly, patient satisfaction levels were not widely discussed in the literature, but there were indications they were generally satisfied with their telemedicine experience, provided they could connect with their providers in the virtual domain and achieve desirable outcomes from the encounters [7,9,27,31]. While many patients prefer to be with their providers in-person for face-to-face encounters, and likewise from the provider perspective, telemedicine was viewed as a suitable replacement considering the present COVID-19 environment.

### 4.2. Limitations

We encountered a series of limitations during our research: (1) Recent phenomenon of COVID-19 dramatically restricts our research timeline and availability of pertinent articles due to novel, mercurial circumstances. In the two-month span of our study, additional articles were likely published after our initial capture that were not employed in our final analysis. (2) With protective measures ebbing and flowing in real time in response to the changing tides of COVID-19′s effects, healthcare services may reinstitute in-person services (i.e., elective surgeries, minor procedures, physical exams, etc.) or scale them back as conditions on the ground dictate in any given locality; these fluctuations, in turn, will cause a see-saw effect of telemedicine utilization. (3) Selection bias—we attempted to control against this variable through a rigorous vetting process via group consensus. (4) Publication bias—our searches were narrowly conducted across only two databases and did not include any forms of grey literature; as a result, it is probable we did not secure supplementary articles that may have illuminated additional information germane to our research. (5) Due to the novel presentation of SARS-CoV-2, source material was generally in short supply as there was a dearth of peer-reviewed studies to inform our search results.

## 5. Conclusions

Telemedicine has had a transformative impact on the provision of care in the era of COVID-19. It is evident from the presented research that the service lines covered are demonstrating nimble and effective responses to the COVID-19 outbreak through workflow adaptations via telemedicine within their respective care provisions. While general obstacles were encountered, which encompassed a lack of reimbursement parity, telemedicine infrastructure capabilities, regulatory and HIPAA compliance guidelines, lack of internet connectivity, and patient/provider discomfort with technology, each developed the capacity to accelerate telemedicine adoption to adjust to the needs of their patient populations by marshaling resources, expertise, and access. Since the SARS-CoV2 pandemic thrust telemedicine into uncharted territory, its capabilities continue to be refined as best practices are codified and datasets are assembled for utilization in future peer-reviewed publication dissemination. It behooves legislative and industry leaders to re-examine the benefits of telemedicine to remove barriers to its application not just in times of public health crises, but also for normal and customary clinical practice.

## Figures and Tables

**Figure 1 healthcare-08-00380-f001:**
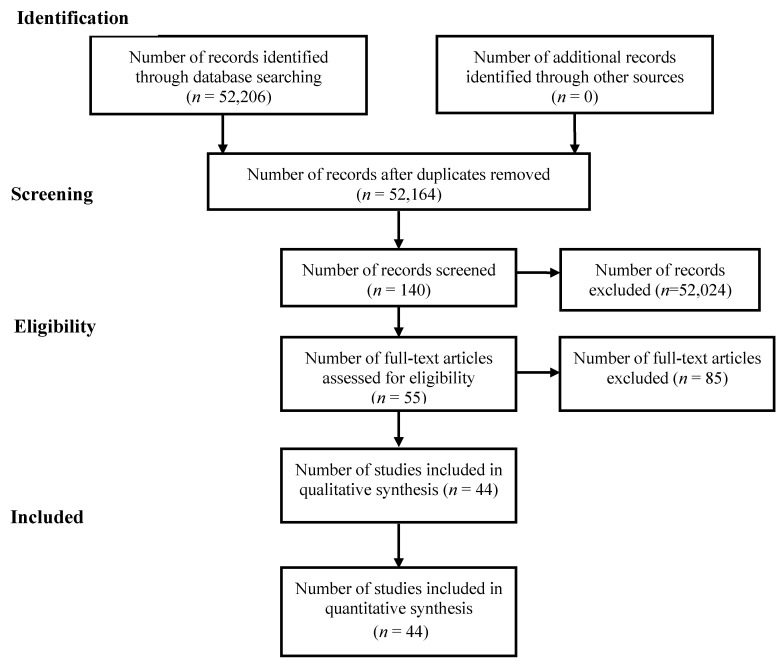
Preferred reporting items for systematic reviews and meta-analyses flow diagram of the literature search and selection process.

**Table 1 healthcare-08-00380-t001:** Use of telemedicine prior to and following the 2020 COVID-19 pandemic.

User	Prior to 2020 COVID-19 Pandemic	Following 2020 COVID-19 Pandemic
**Patients**	9.6% of the population used telemedicine for clinical encounters 74.3% noted they either did not have access or were unaware of telemedicine options [9].	Recent study showcased a 14% surge in telehealth visits in a given week across a two-month stretch from mid-March 2020 to mid-May 2020 [10].
**Providers**	18% of physicians provided a telemedicine platform for their patients in 2018 [11].	Physician practice deployment of telemedicine usage mushroomed to 48% as healthcare providers scrambled to minimize gaps in care provision amid the SARS-CoV-2 global pandemic [12].

**Table 2 healthcare-08-00380-t002:** Summary of results.

First Author, Date, Reference	Title	Publication	Observations	Themes	Service Line Addressed
Ludwig, 2020 [8]	Coronaviruses and SARS-CoV-2: A Brief Overview	Anesthesia and Analgesia	Several cases of pneumonia of unknown origin were reported from China, which in early January 2020 were announced to be caused by a novel coronavirus. The virus was later denominated severe acute respiratory syndrome coronavirus 2 (SARS-CoV-2) and defined as the causal agent of Coronavirus Disease 2019 (COVID-19)		Infectious Disease
J.D. Power, 2019 [9]	One in 10 Americans Use Telehealth, But Nearly 75% Lack Awareness or Access, J.D. Power Finds	PR Newswire	While 9.6% of Americans have used telehealth services, nearly three-fourths (74.3%) say they either do not have access or are unaware of telehealth options		Multiple Specialties
Eddy, 2020 [10]	Nearly half of physicians using telehealth, up from just 18% in 2018	Healthcare IT News	Physicians are changing the patterns of their practice because of the COVID-19 pandemic, with nearly half of them using telehealth to treat patients, up from just 18% in 2018		Multiple Specialties
Lehrman, 2020 [13]	Telemedicine Options: The COVID-19 pandemic underscores the role of remote patient management	Podiatry Management	Certain telemedicine services can be provided year-round Rapid acceleration of telehealth since the advent of COVID-19, enforcement discretion and penalty waivers were issued for HIPAA violations while providing these services	HIPAA compliance; rapid expansion	Multiple Specialties
Gondal, 2020 [14]	Telemedicine in the time of COVID-19 Pandemic	Journal of the College of Physicians and Surgeons	Disadvantages of telemedicine is that physical examination, laboratory, or radiological tests cannot be performed virtually Lack of education, communication network availability and awareness also hinders its acceptance by the people at large Telemedicine may be used for medical education and distant learning/teaching	“Telemedicine education; risk mitigation; convenience; lower cost; social distancing promotion; improved access”	Multiple Specialties
Reynolds, 2020 [15]	Telehealth in Pregnancy (Editorial)	Lancet Diabetes and Endocrinology	In the rapid implementation of delivery of remote antenatal care in response to COVID-19 there remain many uncertainties There is limited knowledge about women’s views of use of telehealth for monitoring pregnancy complications, although available data suggests that women find this to be a positive experience There is concern that most of the trials of telehealth technologies have been done in highly selected groups and so the findings might not be applicable to the wider population Gestational weight gain and optimal wellness (GLOW) was a randomized trial of a weight management intervention delivered by telephone during a pregnancy with an aim of reducing gestational weight gain in women with overweight or obesity	Social distancing; Self-Isolation; Risk mitigation	OB/GYN OB/GYN
Machado, 2020 [16]	Social Media and Telemedicine for Oral Diagnosis and Counselling in the COVID-19 Era (Editorial)	Oral Oncology	“Forward triaging” is possible to screen for SARS-COV2–2 symptoms and mitigate community spread Alternative forms of telemedicine usage (e.g., text messaging, e-mailing, social media messaging) are encouraged to avoid long lines, rule out oral lesions, and obtain early diagnoses	Social distancing; Self-Isolation; Risk mitigation	Dentistry
AHC Media, 2020 [17]	COVID-19 Shuts Down Nation; Family Planning Need Not Stop: Clinics resort to remote care.	Relias Media	Clinics can phone triage patients before a scheduled visit to determine whether the visit can be done by telephone visit, or synchronous or asynchronous telemedicine Medicaid and other payers could cover telehealth services	“Improved access; increased reimbursements; social distancing; risk mitigation; telemedicine education”	OB/GYN
Canady, 2020 [18]	COVID-19 outbreak represents a new way of mental health service delivery. (Editorial)	Mental Health Weekly	Staff are equipped with iPads and other equipment necessary to work from home Staff are provided training Medications can be left at patient’s door	“Telemedicine education; Social distancing promotion; improved access”	Mental Health
Ohannessian, 2020 [19]	Global Telemedicine Implementation and Integration Within Health Systems to Fight the COVID-19 Pandemic: A Call to Action (Editorial)	JMIR Public Health and Surveillance	Telemedicine shown to be helpful in previous outbreaks Two examples of telehealth include (1) direct-to-consumer telemedicine with private providers mostly relying on out-of-pocket or private insurance payment, and (2) free solutions, mainly from US-based companies (for example, WhatsApp, Skype, or Facetime), that may not respect national health data privacy and security requirements. A scientific evaluation framework and dedicated research funds to describe and assess the impact of telemedicine during outbreaks	“Improved access; telemedicine education”	Multiple Specialties
Lowery, 2020 [20]	Telehealth: A new frontier in OB/GYN	Contemporary OB/GYN	In utilizing telehealth, the overall healthcare system benefits from lower costs, less travel, improved health outcomes, and reduced emergency room utilization Barriers include: Licensing and credentialing, malpractice insurance, and reimbursement and billing	“Social distancing; convenience; lower costs; risk mitigation”	OB/GYN
AHC Media, 2020 [21]	COVID-19 Devastates At-Risk Populations: Telemedicine could be new normal	Relias Media	Case managers should focus more on remote case management, taking the pandemic into account as they contact and monitor patients	Risk mitigation; social distancing	Multiple Specialties
Romanick-Schmied, 2020 [22]	Telemedicine Maintaining Quality During Times of Transition	Nature Reviews Disease Primers	The COVID-19 crisis has accelerated the adoption of telemedicine Clinics and hospitals have the obligation to communicate to patients that all possible means are being taken to prevent transmission of infection while maintaining quality in the delivery of care The advantage of convenience from conducting a telemedicine visit must be balanced and weighed against the benefits of direct human interactions	Improved access; rapid expansion; social distancing; convenience; risk mitigation	Multiple Specialties
Vidal-Alaball, 2020 [23]	Telemedicine in the Face of the COVID-19 Pandemic	Atencion Primaria	Telemedicine connects the convenience, low cost, and ready accessibility of health-related information and communication using the Internet and associated technologies Telemedicine during the coronavirus epidemic has been the doctors’ first line of defense to slow the spread of the coronavirus, keeping social distancing and providing services by phone or videoconferencing	Improved access; rapid expansion; social distancing; lower cost; convenience	Multiple Specialties
Wosik, 2020 [24]	Telehealth transformation: COVID-19 and the rise of virtual care	Journal of the American Medical Informatics Association	The pandemic has catalyzed rapid adoption of telehealth, or the entire spectrum of activities used to deliver care at a distance Healthcare enterprises may already have in place technologies that can be employed to accomplish telehealth	Rapid expansion; social distancing	Multiple Specialties
Bashshur, 2020 [25]	Telemedicine and the COVID-19 Pandemic, Lessons for the Future (Editorial)	Telemedicine Journal and e-health	Conversion to telemedicine demonstrates its utility as an effective tool for social distancing A sizeable proportion of outpatient visits can be clinically managed effectively from a distance Government has relaxed all restrictive regulations for telemedicine deployment. The necessary logistics can be developed promptly	“Improved access; increased reimbursement; rapid expansion; social distancing; convenience; lower cost; risk mitigation; telemedicine education; relaxed regulations”	Multiple Specialties
Spinelli, 2020 [26]	COVID-19 Pandemic: Perspectives on an Unfolding Crisis	British Journal of Surgery	Telemedicine may reduce the need for physical attendance in outpatient clinics, minimizing contact exposure Surgical staff and the available units have been modified to balance service provision, reducing infection risk, and specialist care	Risk mitigation; social distancing	Multiple Specialties
Mann, 2020 [27]	COVID-19 Transforms Health Care Through Telemedicine: Evidence From the Field	Journal of the American Medical Informatics Association	U.S. insurers have quickly expanded coverage to include all telemedicine visit types The U.S. Department of Health & Human Services (HHS) waived enforcement of HIPAA regulations to allow the use of consumer audio and video communication for telemedicine visits	HIPAA compliance; increased reimbursement; rapid expansion; social distancing; patient satisfaction; risk mitigation; telemedicine education	Multiple Specialties
Aziz, 2020 [28]	Telehealth for High-Risk Pregnancies in the Setting of the COVID-19 Pandemic	American Journal of Perinatology	Telehealth for pre-natal care is feasible Tailored regimens for increased surveillance and counseling are permissible Certain high-risk pregnancies may require increased frequency of in-person encounters	Mitigate risk for COVID-19 exposure; minimize patient travel	OB/GYN
Zhou, 2020 [29]	The Role of Telehealth in Reducing the Mental Health Burden from COVID-19 (Editorial)	Telemedicine Journal and e-health	Tele-mental health services are feasible, appropriate, and perfectly suited to the current pandemic environment Simple communication methods (e.g., e-mails, texts, etc.) should be used more extensively	Increased access; infection risk mitigation	Mental Health
Freeman, 2020 [30]	COVID-19 From a Psychiatry Perspective: Meeting the Challenges	The Journal of Clinical Psychiatry	We have seen telemedicine set up at record speed to meet the needs of patients Regulatory barriers to reach many patients were brought down almost overnight	Regulatory barrier removal, rapid expansion, improved access	Mental Health
AHC Media, 2020 [31]	Shift to Telehealth Could Remain Trend After COVID-19: Reproductive health remains priority.	Relias Media	Clinics have quickly shifted to phone screenings and videoconferences No penalties for Health Insurance Portability and Accountability Act (HIPAA) noncompliance Positive experience from patients and providers Medicare waiving copays and deductibles	HIPAA compliance; lower costs; patient satisfaction	OB/GYN
Knoph, 2020 [32]	Telepsychiatry Coming Into Its Own With COVID-19 (Editorial)	Brown University Child and Adolescent Pharmacology Update	Children find tele-psychiatry easy to use since they are used to technology Children with anxiety or trauma feel more comfortable with tele-psychiatry Increase in patient and provider satisfaction rate	“Patient satisfaction; reimbursement; improved access”	Mental Health
Parikh, 2020 [33]	Cardio-Oncology Care In the Time of COVID-19 and the Role of Telehealth	JACC CardioOncology	Many cardio-oncology visits cannot be safely deferred Officials have approved interstate licensing Patients can be monitored remotely	“Improved access; risk mitigation; convenience”	Oncology
Rodler, 2020 [34]	Lessons from the coronavirus disease 2019 pandemic: Will virtual patient management reshape uro-oncology in Germany?	European Journal of Cancer	Patients with exposure to COVID follow up via phone or e-mail and asked to write a symptom diary Virtual management and reductions in frequency of visits are feasible	Risk mitigation	Oncology
Steingass, 2020 [35]	Telehealth Triage and Oncology Nursing Practice	Seminars in Oncology Nursing	Oncology nurses must understand various methods of telehealth and how to establish and work in the new environment Organization must ensure that the telehealth program is focused on meeting the clinical needs of the defined patient population Identifying ways to improve gaps in care coordination or management will help determine what types of telehealth encounters must be deployed within the organization	Telemedicine education	Oncology
AHC Media, 2020 [36]	Hospitals Use Telemedicine to Limit Exposures, Preserve Personal Protective Equipment (PPE), Guide Patients to Right Setting	Relias Media	Centers for Medicare & Medicaid Services (CMS) pledges penalties will not be imposed on providers who use telehealth in ways that are not compliant with HIPAA requirements Clinicians can bill for telemedicine visits with reimbursement rates on par with in-person visits	“Improved access; HIPAA compliance; increased reimbursement; rapid expansion”	Multiple Specialties
Bokolo, 2020 [37]	Use of Telemedicine and Virtual Care for Remote Treatment in Response to COVID-19 Pandemic	Journal of Medical Systems	Hospitals can improve the efficiency of the medical system by replacing a proportion of physical treatments with digital technologies CMS now allow medical-care providers to utilize devices such as smartphones and electronic devices to treat patients The US drug enforcement administration is also allowing medical practitioners to prescribe of medication after patient diagnosis and assessment conducted via telemedicine	Rapid expansion; risk mitigation	Multiple Specialties
Portnoy, 2020 [38]	Telemedicine in the Era of COVID-19 (Editorial)	The Journal of Allergy and Clinical Immunology	Telemedicine (TM) has the potential to help by permitting mildly ill patients to get the supportive care they need while minimizing their exposure to other acutely ill patients Nearly all health plans and large employers offer some form of coverage for TM services Patients may be unaware that they have TM as an option and do not know how to access it For routine telemedicine video visits, many of the requirements have been waived during this time	“Risk mitigation; improved access; increased reimbursement; HIPAA compliance”	Multiple Specialties
Contreras, 2020 [39]	Telemedicine: Patient-Provider Clinical Engagement During the COVID-19 Pandemic and Beyond	Journal of Gastrointestinal Surgery	In response to the COVID-19 pandemic, federal agencies have promoted telehealth both through regulatory relaxation and increased funding Telemedicine can facilitate an international reach to patients across the world	Increased reimbursement; relaxed regulations; improved access	Multiple Specialties
Boxer, 2019 [40]	Advantages and utilization of telemedicine	mHealth	Authors explore the various reasons for telemedicine utilization and offer suggestions to increase the delivery of high-quality telemedicine services; telemedicine is, at present, a primary means of expanding care to those with limited access to physicians. Some states have introduced cross-state licensing Economics is another barrier to the expansion of telemedicine. Congressional Telehealth Caucus reintroduced several bills aimed at addressing Medicare barriers impeding telemedicine efforts, these include an expansion of the scope of reimbursed services along with redefining rural qualifications	Improved access; relaxed regulations; increased reimbursement	Multiple Specialties
Jobes, 2020 [41]	The COVID-19 pandemic and treating suicidal risk: The telepsychotherapy use of CAMS	Journal of Psychotherapy	The pandemic has thus sparked a sudden interest in providing mental health services via tele-psychotherapy The coronavirus pandemic now poses all new approval concerns about the routine practice of having an acutely suicidal patient go to an emergency department and/or admitting such patients to an inpatient psychiatric unit	Telemedicine education	Mental Health
Elkaddoum, 2020 [42]	Telemedicine for cancer patients during COVID-19 pandemic: between threats and opportunities	Future Oncology	It was clearly recommended to switch to telemedicine as much as possible for patients who present new symptoms or side effects, despite being considered high- to medium-priority patients This approach minimizes the need for individuals to visit healthcare facilities, leading to a lesser consumption of PPE by the patients and the doctors	Improved access; risk mitigation	Oncology
Trinidad, 2020 [43]	Telemedicine for inpatient dermatology consultations in response to the COVID-19 pandemic (Editorial)	Journal of the American Academy of Dermatology	Dermatologists are poised to use tele-dermatology to increase access to dermatologic care for hospitalized patients, reduce the risk of infection of patients, trainees, and staff, and reduce the use of precious resources such as personal protective equipment and medical supplies	Improved access; risk mitigation	Dermatology
Lovett, 2020 [44]	Incorporating Telemedicine as part of COVID-19 Outbreak Response Systems	The American Journal of Managed Care	Telemedicine platforms are ideal for managing several challenges facing healthcare systems in response to global infectious disease outbreaks Implementing telemedicine systems focuses on addressing the needs of low-acuity patients with disease exposure concern State and federal laws and regulations have evolved in recent years, months, and days to facilitate greater reimbursement for and adoption of various telemedicine models. State, federal, and international laws and regulations have been significantly relaxed to promote greater adoption and use of telemedicine and other digital health technologies to deliver clinical services	“Improved access; increased reimbursement; rapid expansion; risk mitigation, relaxed regulations”	Multiple Specialties
Goldberg, 2020 [45]	Tele-dermatology and HIPAA compliance in the era of COVID-19.	Dermatology Times	There exist Liability issues under the HIPAA that physicians and dermatologists will be facing when using telemedicine or tele-dermatology to communicate with patients amidst the coronavirus pandemic These include being subjected to disciplinary action if they exceed the license granted by his/her state, and negligence liability for doctors providing the telemedicine service in a state different from where the patient lives E-health and Telemedicine are great tools that can be utilized within the healthcare industry especially during the COVID pandemic	HIPAA compliance	Dermatology
Prasad, 2020 [46]	Optimizing Your Telemedicine Visit During the COVID-19 Pandemic: Practice Guidelines for Patients With Head and Neck Cancer	Head and Neck	Guidelines are necessary to set patient expectations and to also ensure both providers and patients are appropriately educated on telemedicine platform functionality	Patient/Provider Telemedicine Education	Oncology
Badeli, 2020 [47]	Utilizing Telemedicine for Managing COVID-19	Journal of Pediatric Nephrology	The integration of eHealth for patients with COVID-19 is indicated as a global emergency to reduce virus transmission Telemedicine provides a promising solution to provide quality care while reducing risk of transmissions to both other patients and healthcare providers Previous investigations on the effect of this method on outpatients with chronic stable diseases showed its cost effectiveness, high satisfaction among patients and caregivers, and no significant difference in service use or disease progression, it is not an appropriate method for all clinical situations. Not appropriate for patients with severe diseases. As Nephrology consultation needs commonly laboratory assessments, video consultation can help patients to receive disease education and explanation of treatment choices	“Improved access; social distancing; lower cost; patient satisfaction”	Oncology
Cosic, 2020 [48]	Impact of Human Disasters and COVID-19 Pandemic on Mental Health: Potential of Digital Psychiatry	Psychiatria Danubina	Comprehensive approach based on digital psychiatry is proposed to address the lack of access to psychiatric services, which includes artificial intelligence, tele-psychiatry and an array of new technologies, like internet-based computer-aided mental health tools and services	Improved access; telemedicine education	Mental Health

**Table 3 healthcare-08-00380-t003:** Affinity matrix.

Effectiveness Themes	Article Citation Number	Incidence of Occurrence (*n* = 43)	Probability of Occurrence
Improved Access	[1,13,17,20,23,24,25,29,30,31,32,33,35,37,38,39,40,41,42,44,47]	21	48%
Risk Mitigation	[1,13,15,16,17,20,21,22,23,25,27,28,29,33,34,36,37,39,43]	19	43%
Social Distancing Promotion	[13,15,16,17,20,21,23,24,27,35,36,37,41,44]	14	32%
Rapid Telemedicine Expansion	[13,22,23,24,25,26,30,32,33,35,37,44]	12	27%
HIPAA Compliance/Relaxed Regulations	[13,25,26,30,31,37,38,39,40,47,48]	11	25%
Telemedicine Education	[13,17,19,20,35,37,38,44,45,46]	10	23%
Increased or Similar Reimbursements	[13,17,25,37,38] [39,40,47]	8	18%
Convenience	[13,20,21,23,24,31,33]	7	16%
Lower Cost	[13,20,21,24,38,41]	6	14%
Patient Satisfaction	[31,32,37,41]	4	9%

**Table 4 healthcare-08-00380-t004:** Service line successes through telemedicine utilization during the COVID-19 pandemic.

Service Line	Implementation Successes
Dermatology	Rapid diagnosis and treatment, increased access, reduced risk of infection, minimization of PPE and medical supply utilization, addressed health care disparities for underserved and rural populations.
OB/GYN	A typical virtual visit includes the pregnant woman utilizing home monitoring supplies to track measures such as fetal heart rate, maternal blood pressure, and fundal height. Specialists are able to view ultrasounds and other examinations through available technology, often in real-time. Home monitoring is also possible for certain high-risk conditions. Clinics quickly shifted to phone screening and initial consultations, as well as videoconferences with patients. Some clinics provided contraception renewals and new prescriptions through telemedicine. From a family planning perspective, telehealth visits have been a positive experience that both patients and providers favor. Clinics can phone triage patients before a scheduled visit to determine whether the visit can be done by telephone visit, or synchronous or asynchronous telemedicine. GLOW was a randomized trial of a weight management intervention delivered by telephone during a pregnancy with an aim of reducing gestational weight gain in women with overweight or obesity—goals of implementation were to consolidate in-person prenatal screening, surveillance, and examinations into fewer in-person visits while maintaining patient access to ongoing antenatal care and subspecialty consultations via telehealth virtual visits.
Oncology	Less consumption of PPE, can ensure patients are adhering to given recommendations, patients can address new symptoms, fears, or questions with providers. Cardio-oncology patients can be evaluated regularly, e.g., blood pressure readings, weight scales, or Cardio Micro Electro Mechanical System (CardioMEMS). Strict quarantine observation, reduced clinic visits, survivorship care planning, patient education.
Mental Health	Supporting both physical and psychosocial needs irrespective of geographic locations. Children with anxiety or significant trauma feel more comfortable than they would with in-person therapy. In response to the sudden need to provide tele-psychotherapy services, organizations quickly developed and offered free resources, clinical guidance, and synchronous and asynchronous access to online presentations to thousands of mental health providers around the world. Prescriptions can now be made (but not for other controlled substances, such as ADHD medications) after a thorough assessment through a live interactive video. Collaborative Assessment and Management of Suicidality (CAMS) protocols were created to provide support, guidance, and resources. Increase in patient and provider satisfaction rates.

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
