# Peer review of "The Impact of COVID-19 on Telemedicine Utilization Across Multiple Service Lines in the United States"

_healthcare, 2020, doi:10.3390/healthcare8040380_

Round 1

Reviewer 1 Report

Abstract

Background is well stated and attention grabbing.

Methods could benefit from more detailed description. Though it is the abstract, the reflection of the methods should still be as descriptive as possible given this is a systematic review.

Likewise, results would be better reflected by more precise detail, e.g. description of participants and intervention characteristics, direction of outcome effect etc. When the reader reaches the conclusion, it is making a large leap from what is presented in the results.

Introduction

90-92: Last sentence of ‘Rationale’ could benefit from some reworking to be clearer.

94-99: The authors my consider putting this information into a box? It is currently just a long list. The authors could develop the picture about challenges telemedicine faced, prior to the ‘accomodations’, a little more.

Significance is well stated. Though the sentence beginning line 105, is extremely long and would benefit from being broken down a little.

What is ‘unique health service lines’? why specifically those disciplines?

Methods

The authors mention following PRISMA principles, but was a review protocol predefined? Is it published anywhere online, like PROSPERO?

How was a tangential relationship to COVID-19 determined in order to exclude potential studies?

131-134: Does this refer to the information presented in the background? Why is this mentioned as part of the methods? It does not relate to the objective of the study. The search that is described in methods should only describe the search that yielded the results.

Eligibility criteria currently reads as a list of search limiters but does not actually define the criteria that made studies eligible for inclusion. What research designs were accepted? What definitions of telemedicine were accepted?...

It is recommended that the authors include more detail about the specific search terms and strategy that they used to conduct the literature search for this review. It is best practice in review reporting to include at least one full example of a search strategy used in a database. Was a Research Librarian involved in helping to determine the most appropriate search strategy?

Did each author review the list of search results independently? Was each record only screened once? What about full-text screening?

146: What do the authors mean by ‘filtered results’?

Search terms were refined weekly? At what stage did this stop? Unsure that this sentence about weekly author meetings is necessary.

149: what is meant by ‘loose ends’?

154-167: this is more of a discussion point than methodology

All well and good to confirm author agreement on data extraction but was there a specific risk of bias assessment undertaken for each study?

Did the authors have any pre-determined outcomes of interest?

Study Selection and Data Collection Process: first paragraph belongs under Results; second paragraph belongs further up under Eligibility Criteria. The latter would have answered some of my earlier questions.

Description of results for this systematic review do not follow the usual reporting for systematic reviews and it would not be a replicable study based on what is presented currently in this manuscript.

Figure 1: why are there two identical boxes reflecting the number of studies in the qualitative synthesis? Reasons for exclusion of full-text articles are missing.

Results

They synthesis of results is very short on the characteristics and findings of the included studies. This makes it a little difficult for the reader to pull out what the significance of the findings of this review are.

Table 1: there is a column. For outcomes but only two studies have an entry for this field, therefore is it necessary?

It would be nice to see here what type of study the publication was e.g. RCT, cross-sectional survey…etc.

Publication dates are all presented in a different format.

I am not clear about the relevance/ importance of including articles that are historical to the objective of the review. That is, articles pre-dating COVID-19?

Discussion

Discussion does not really develop or discuss ideas in relation to the wider scientific literature. It appears to be more of a summary of the main results, in fact something like this would have been beneficial in the results section.

Nice to see the limitations of the review stated. Would be excellent to also see if the authors attempted to address these in anyway through their methodology.

Conclusion

Conclusion is interesting but it was difficult to glean this from reading the results section as it is currently presented.

Overall

Unsure that this topic and available evidence base is well suited to the systematic review format. The authors may have more freedom to comment and develop ideas based on the limited literature in a less structured review method?

The presentation of this study as it is, is quite disjointed and lacking detail to substantiate its purpose. Reporting of methods relating to systematic review practice is lacking.

Author Response

Thank you for giving us the opportunity to submit a revised draft of the manuscript, “The Impact of COVID-19 on Telemedicine Utilization Across Multiple Service Lines“ for publication in Healthcare. We appreciate the time and effort that you and your reviewers dedicated to providing feedback on our manuscript and are grateful for the insightful comments on the valuable improvements to our paper. We have incorporated the recommendations made by the reviewers. These changes are highlighted within the manuscript for your ease in reading. Please see below (in blue text), for a point-by-point response to the reviewers’ comments and concerns. All page numbers refer to the revised manuscript file with tracked changes.

Reviewers’ Comments to the Authors:

Reviewer 1

Reviewer 1 comment: Abstract:

  1. Background is well stated and attention grabbing.
  2. Methods could benefit from more detailed description. Though it is the abstract, the reflection of the methods should still be as descriptive as possible given this is a systematic review.
  3. Likewise, results would be better reflected by more precise detail, e.g. description of participants and intervention characteristics, direction of outcome effect etc. When the reader reaches the conclusion, it is making a large leap from what is presented in the results.

Author response: Thank you so much taking the time to review our article. We hope that this revision improves the overall content of the first draft.

  1. a) Thank you as it was our sincere intent to capture the interest of the Reader from the very beginning.
  2. b) The study team has revised this section to include a more detailed description of the study methodology.
  3. c) The study team has revised this section to include a more detailed description of the study results.

Reviewer 1 comment: Introduction:

  1. 90-92: Last sentence of ‘Rationale’ could benefit from some reworking to be clearer.
  2. 94-99: The authors my consider putting this information into a box? It is currently just a long list. The authors could develop the picture about challenges telemedicine faced, prior to the ‘accomodations’, a little more.
  3. Significance is well stated. Though the sentence beginning line 105, is extremely long and would benefit from being broken down a little.
  4. What is ‘unique health service lines’? why specifically those disciplines?

Author response:

  1. a) Lines 90-92: The ‘Rationale’ section has been rewritten to be more clear and understandable to the Reader.
  2. b) Lines 94-99: Thank you for the very valuable suggestion of placing the information that was originally in the paragraph, into a table (now found in Table 1).
  3. c) Line 105: sentence has been rewritten for more clarity and conciseness (now begins with line 118 in revised manuscript).
  4. d) Our study team selected the four health service lines of mental health, oncology, OB/GYN and dermatology because we found that these medical specialities encountered similar, surmountable obstacles in their care delivery models. However, each specialty was able to adapt accordingly to the respective needs of their patient populations throughout the COVID-19 pandemic.

Reviewer 1 comment. Methods

  1. The authors mention following PRISMA principles, but was a review protocol predefined? Is it published anywhere online, like PROSPERO?
  2. How was a tangential relationship to COVID-19 determined in order to exclude potential studies?
  3. 131-134: Does this refer to the information presented in the background? Why is this mentioned as part of the methods? It does not relate to the objective of the study. The search that is described in methods should only describe the search that yielded the results.
  4. Eligibility criteria currently reads as a list of search limiters but does not actually define the criteria that made studies eligible for inclusion. What research designs were accepted? What definitions of telemedicine were accepted?...
  5. It is recommended that the authors include more detail about the specific search terms and strategy that they used to conduct the literature search for this review. It is best practice in review reporting to include at least one full example of a search strategy used in a database. Was a Research Librarian involved in helping to determine the most appropriate search strategy?
  6. Did each author review the list of search results independently? Was each record only screened once? What about full-text screening?
  7. 146: What do the authors mean by ‘filtered results’?
  8. Search terms were refined weekly? At what stage did this stop? Unsure that this sentence about weekly author meetings is necessary.
  9. 149: what is meant by ‘loose ends’?
  10. 154-167: this is more of a discussion point than methodology
  11. All well and good to confirm author agreement on data extraction but was there a specific risk of bias assessment undertaken for each study?
  12. Did the authors have any pre-determined outcomes of interest?
  13. Study Selection and Data Collection Process: first paragraph belongs under Results; second paragraph belongs further up under Eligibility Criteria. The latter would have answered some of my earlier questions.
  14. Description of results for this systematic review do not follow the usual reporting for systematic reviews and it would not be a replicable study based on what is presented currently in this manuscript.
  15. Figure 1: why are there two identical boxes reflecting the number of studies in the qualitative synthesis? Reasons for exclusion of full-text articles are missing.

Author response:

  1. a) The study team describe the study review protocol beginning with line 149 in the revised manuscript. Researchers utilized the evidence-based PRISMA principles however found it challenging as COVID-19 repository of information is still developing.
  2. b) Our study team focused on articles specific to healthcare service lines that were directly impacted by the COVID-19 pandemic (e.g. caused them to alter delivery). If the current pandemic did not directly impact healthcare delivery (e.g. clinical procedure involved), then the article was excluded.
  3. c) Original lines 131-134 (149-167 in the revised manuscript): Yes, this sentence referred to the information presented in the background. It is mentioned in this section in order to explain why these article detailing the histocial evolution of telehealth were included. However, if the Reviewers think that we should either delete it or move it to another section we would be happy to do so. Thank you.
  4. d) Thank you for this comment. The eligibility criteria are found in the PRISMA flowchart. Additionally, we describe eligibility criteria in the revised lines 149-167. However, if this is insufficient we can certainly revise with additional details. All research designs were evaluated and accepted. Our study team evaluated the various means of providing telehealth and accepted all studies with a stated link between the current COVID-19 pandemic and the use of telehealth.
  5. e) The study team has inserted in the revised manuscript (lines 144-156) additional details regarding specific search terms and strategies used to conduct the literature search for this review. Unfortunately a research librarian was not utilized during the conduct of this systematic review: an error that will not be repeated in the future.
  6. f) Yes, each study team member reviewed the list of search results independently. Each record was screened at least twice including full-text screening.
  7. g) Line 146 (line 179 in revised manuscript): We have rewritten this sentence and removed the word ‘filtered” for better clarity.
  8. h) Yes, study search term listing was refined weekly as team members met weekly to compare notes as to what they were discovering in their article reviews. This came to an end as our team narrowed the focus of our study and began repeating search term results in their queries.
  9. i) Line 149 (line 182 in revised manuscript): We have rewritten this sentence and removed the term ‘loose ends’ for better clarity.
  10. j) Line 154-167: Our study team concurs with the Reviewer’s recommendation. We have moved this sentence from the methodology section to the discussion section.
  11. k) Thank you for your comment on our study team’s confirmation on author agreement on data extraction as it was important to establish in the article. However, a specific risk of bias assessment was not undertaken for each study. The best we can surmise is that each author in the assessed articles may have been biased towards own medical specialty based on their respective professional knowledge and experience.
  12. l) The study authors did not have any pre-determined outcomes of interest per se, other than we wanted to highlight how the current COVID-19 pandemic has served as a catalyst to accelerate the use of telemedicine by both the Provider community as well as the Patient community in certain healthcare service lines.
  13. m) As recommended by the Reviewer, we have moved the first paragraph under Study Selection and Data Collection Process section to the Results section. This reads much more logically and we are grateful to the Reviewer for their recommendation. Additionally, we move moved the second paragraph under the Eligibility Criteria section for a more logical flow. We are hopeful that these relocations of sections within the paper indeed address some of the Reviewer’s earlier questions.
  14. n) It is unclear to the study team what the Reviewer specifically refers to in terms of our study “not following the usual reporting for reviews. Our study team utilized the evidence-based Preferred Reporting Items for Systematic Reviews and Meta-Analyses (PRISMA) methodology to guarantee a transparent and comprehensive reporting of results. Further elaboration from the Reviewer is requested. Thank you.
  15. o) Figure 1: The Reviewer is correct. We have replaced Figure 1 with a corrected version of the figure. Inclusion criteria for full-text articles are the same as overall inclusion criteria.

Reviewer 1 comment: Results

  1. They synthesis of results is very short on the characteristics and findings of the included studies. This makes it a little difficult for the reader to pull out what the significance of the findings of this review are.
  2. Table 1: there is a column. For outcomes but only two studies have an entry for this field, therefore is it necessary?
  3. It would be nice to see here what type of study the publication was e.g. RCT, cross-sectional survey…etc.
  4. Publication dates are all presented in a different format.
  5. I am not clear about the relevance/ importance of including articles that are historical to the objective of the review. That is, articles pre-dating COVID-19?

Author response:

  1. a) The study team pulled key characteristics and findings of the included studies into the Results section. However, we would be happy to include more specifics if the Reviewer can elaborate further on their recommendation. Thank you.
  2. b) We have renamed the final column in Table 1 to inform the Reader of the primary medical specialty addressed in the article.
  3. c) Thank for this recommendation. We have reformatted the table to include more information that will make more relevant information clearer to the Reader.
  4. d) The date formats in Table 1 have been standardized to read month/date/year (e.g. 4/16/2020).
  5. e) Our study included a number of historical articles in order to provide context for the evolution of telemedicine as the Reviewer correctly noted, has been in existence for quite some time now. Although it is outside the customary systematic review search parameters since some articles pre-date our 2016-2020 search window, it was the intent of the study to select articles that provided a brief history of this evolution. However, if the Reviewer recommends elimination of these historical articles, the study team can easily remove them.

Reviewer 1 comment: Discussion

  1. Discussion does not really develop or discuss ideas in relation to the wider scientific literature. It appears to be more of a summary of the main results, in fact something like this would have been beneficial in the results section.
  2. Nice to see the limitations of the review stated. Would be excellent to also see if the authors attempted to address these in anyway through their methodology.

Author response:

  1. a) While we do not disagree with the Reviewer’s comments, our study team would like more details on what the Reviewer is specifically recommending. Once we have a better understanding of the Reviewer’s comments, we can certainly address. Thank you.
  2. b) Our study team appreciates the Reviewer’s comments on the limitations section. Unfortunately we were unable to devise a means to address these limitations through our methodology.

Reviewer 1 comment: Conclusion

Conclusion is interesting but it was difficult to glean this from reading the results section as it is currently presented.

Overall

  1. Unsure that this topic and available evidence base is well suited to the systematic review format. The authors may have more freedom to comment and develop ideas based on the limited literature in a less structured review method?
  2. The presentation of this study as it is, is quite disjointed and lacking detail to substantiate its purpose. Reporting of methods relating to systematic review practice is lacking.

Author response:

  1. a) The Reviewer is correct that the available evidence base on the chosen topic presents a challenge for the conduct of a systematic review. However, our study team is hopeful that the revised manuscript will better communicate the value of the study findings to the Reader.
  2. b) While the study team is disappointed in the Reviewer’s first assessment of the proposed article, we are hopeful that the revised manuscript will better communicate the value of the study findings to the Reader.

Reviewer 2 Report

Sir, firstly thank you for the opportunity to review this interesting paper on the role of Telemedicine during the ongoing COVID-19 pandemic.

Despite the potential interest to the readers of Healthcare, I think that the paper, in its current version, suffers from several shortcomings that collectively impair its eventual publication.

More precisely:

1) it is rather unclear how the screening procedure was actually performed: methods section reports: "To be included in the study, articles must have explored the effects of COVID-19 on telemedicine utilization to specific service lines in healthcare as chosen by the student researchers, which included dermatology, oncology, obstetrics/gynecology, and mental health". Moreover, while the comparison with pre-COVID era is useful, it is unclear how Authors have dealth with such data. 

2) A systematic review uses systematic methods to collect secondary data, critically appraise research studies, and synthesize findings qualitatively or quantitatively. In the main section of the results, there is no actual summary nor synthesis of the results, that are summarized in an "additional analysis" that is introduced without a clear reporting of the methods. 

Moreover, there are several minor issues that still impact on the overall merit of the paper:

1) Authors introduce their research reporting about the Ancient Greeks/Romans who "transmitted messages through the application of crude
mediums such as smoke signals and light reflection to announce plague outbreaks and health events such as births and deaths". Despite the cultural interest, and the older myself who studied the original greek test of the Sophoclean Tragedies rejoyced of such quotation, this has nearly nothing to do with the working definition of telemedicine reported by the very same authors, that is: "remote provision of health care services using technology to exchange information for the diagnosis, treatment, and prevention of disease.” Please remove it.

2) Author gave to their introduction a very US-focused approach, but they should be aware that telemedicine was extensively used in countries other than the US well before the quoted reports, for example Australia and Lesser Islands of Europe and Oceania (e.g.https://www.who.int/goe/policies/countries/aus__support_tele.pdf https://www.researchgate.net/publication/224067738_Telemedicine-islands_project_cost-effectiveness_and_cost-comparison_analysis ), and such experiences should be properly cited.

3) Table 1: First column is incosistent, as dates are reported sometimes as MM/DD/YYYY and sometimes as DD-MMM; second/third/fourth columns may be omitted or simplified (e.g. --> FIRST AUTHOR'S NAME et al); Fifth column should be systematized and sixth column as well (the latter one, by identifying some specific themes)

In summary, despite the potential interest, Authors should be aware that an extensive overhaul of their research will be stricly necessary before a full publication.

Author Response

Thank you for giving us the opportunity to submit a revised draft of the manuscript, “The Impact of COVID-19 on Telemedicine Utilization Across Multiple Service Lines“ for publication in Healthcare. We appreciate the time and effort that you and your reviewers dedicated to providing feedback on our manuscript and are grateful for the insightful comments on the valuable improvements to our paper. We have incorporated the recommendations made by the reviewers. These changes are highlighted within the manuscript for your ease in reading. Please see below (in blue text), for a point-by-point response to the reviewers’ comments and concerns. All page numbers refer to the revised manuscript file with tracked changes.

Reviewers’ Comments to the Authors:

Reviewer 2

Reviewer 2 comment:

a) Despite the potential interest to the readers of Healthcare, I think that the paper, in its current version, suffers from several shortcomings that collectively impair its eventual publication.

b) it is rather unclear how the screening procedure was actually performed: methods section reports: "To be included in the study, articles must have explored the effects of COVID-19 on telemedicine utilization to specific service lines in healthcare as chosen by the student researchers, which included dermatology, oncology, obstetrics/gynecology, and mental health". Moreover, while the comparison with pre-COVID era is useful, it is unclear how Authors have dealth with such data. 

c) A systematic review uses systematic methods to collect secondary data, critically appraise research studies, and synthesize findings qualitatively or quantitatively. In the main section of the results, there is no actual summary nor synthesis of the results, that are summarized in an "additional analysis" that is introduced without a clear reporting of the methods. 

d) Authors introduce their research reporting about the Ancient Greeks/Romans who "transmitted messages through the application of crude mediums such as smoke signals and light reflection to announce plague outbreaks and health events such as births and deaths". Despite the cultural interest, and the older myself who studied the original greek test of the Sophoclean Tragedies rejoyced of such quotation, this has nearly nothing to do with the working definition of telemedicine reported by the very same authors, that is: "remote provision of health care services using technology to exchange information for the diagnosis, treatment, and prevention of disease.” Please remove it.

e) Author gave to their introduction a very US-focused approach, but they should be aware that telemedicine was extensively used in countries other than the US well before the quoted reports, for example Australia and Lesser Islands of Europe and Oceania (e.g.https://www.who.int/goe/policies/countries/aus__support_tele.pdf https://www.researchgate.net/publication/224067738_Telemedicine-islands_project_cost-effectiveness_and_cost-comparison_analysis ), and such experiences should be properly cited.

f) Table 1: First column is incosistent, as dates are reported sometimes as MM/DD/YYYY and sometimes as DD-MMM; second/third/fourth columns may be omitted or simplified (e.g. --> FIRST AUTHOR'S NAME et al); Fifth column should be systematized and sixth column as well (the latter one, by identifying some specific themes)

g) In summary, despite the potential interest, Authors should be aware that an extensive overhaul of their research will be stricly necessary before a full publication.

Author response:

a) While the study team is disappointed in the Reviewer’s first assessment of the proposed article, we are hopeful that the revised manuscript will better communicate the value of the study findings to the Reader.

b) Thank you for this comment. The eligibility criteria are found in the PRISMA flowchart. Additionally, we describe eligibility criteria in the revised lines 142-153. However, if this is insufficient we can certainly revise with additional details. The study team has inserted in the revised manuscript (lines 144-156) additional details regarding specific search terms and strategies used to conduct the literature search for this review.

c) We are grateful for the Reviewers comments however our study team highlights the results in lines 232 to 257 and additionally in Table 2 of the revised manuscript. However, we would most certainly welcome specific recommendations from the Reviewer of what we have missed. Thank you.

d) While we appreciate the Reviewer’s knowledge of the ancient Greeks and Romans, it was our intent to highlight the fact hat telemedine has a historical context which cannot be dismissed in this paragraph.

e) The Reviewer is absolutely correct in their statement that telemedicine was extensively used in countries other than the US well before the quoted reports. However, in an attempt to focus the scope of this study, the study team chose to only include US-based telehealth initiatives.

f) Table 1 has been standardized by First author, year of publication and reference number. The original fifth was systematized and the final column now lists the medical specialty addressed in the article.

g) While the study team is disappointed in the Reviewer’s first assessment of the proposed article, we are hopeful that the revised manuscript will better communicate the value of the study findings to the Reader. Thank you!

Reviewer 3 Report

The authors present a systematic literature review of the impact of COVID-19 on provision of care through telemedicine. They focus on several specialties/health service lines that were chosen for their uniqueness and that a lot had been written in the literature about this topic for these specialities. The authors identify common themes in these publications. The articles identified were from several different countries, indicating the global impact of COVID-19. Overall a good summary of the current state of the literature as of June 2020.

Major comments

  1. Scope of article- The introduction comes across as wanting to be US focused, however a significant number of articles identified and included in the review do not focus on the US healthcare setting.
  2. Materials and methods – eligibility criteria -Why have the publications been chosen from 2016 to 2020 instead of 2019 to 2020 if the aim is to assess the impact of COVID-19 on telemedicine utilisation?
  3. Table 1. The date formats are not consistent across all rows in table 1.

For example – “20-Jun”, “4/16/20”, “1-May-20”

  1. How is the table 1 ordered? By date, article name, author name or speciality? The order in the table does not seem to match the order in the reference list either.
  2. I think it would help the reader if Table 1 had an additional column describing which service line the article focused on.

Author Response

Thank you for giving us the opportunity to submit a revised draft of the manuscript, “The Impact of COVID-19 on Telemedicine Utilization Across Multiple Service Lines“ for publication in Healthcare. We appreciate the time and effort that you and your reviewers dedicated to providing feedback on our manuscript and are grateful for the insightful comments on the valuable improvements to our paper. We have incorporated the recommendations made by the reviewers. These changes are highlighted within the manuscript for your ease in reading. Please see below (in blue text), for a point-by-point response to the reviewers’ comments and concerns. All page numbers refer to the revised manuscript file with tracked changes.

Reviewers’ Comments to the Authors:

Reviewer 3

Reviewer 3 comment:

a) Scope of article- The introduction comes across as wanting to be US focused, however a significant number of articles identified and included in the review do not focus on the US healthcare setting.

b) Materials and methods – eligibility criteria -Why have the publications been chosen from 2016 to 2020 instead of 2019 to 2020 if the aim is to assess the impact of COVID-19 on telemedicine utilisation?

c) Table 1. The date formats are not consistent across all rows in table 1. For example – “20-Jun”, “4/16/20”, “1-May-20”

d) How is the table 1 ordered? By date, article name, author name or speciality? The order in the table does not seem to match the order in the reference list either.

e) I think it would help the reader if Table 1 had an additional column describing which service line the article focused on.

Author response:

a) The Reviewer is correct in their statement that telemedicine was extensively used in countries other than the US. However, in an attempt to focus the scope of this study, the study team chose to mainly focus on US-based telehealth initiatives while describing systems found internationally.

b) The study team’s intent was to highlight the historical context of telehealth in this manner.

c) The date formats in Table 1 have been standardized to read month/date/year (e.g. 4/16/2020).

d) Table 1 has been reformatted for better clarity in terms of author, date, and specific medical specialty addressed in the article.

e) The final column in Table 1 now describes which service line the article focused on. Again, thank you and your reviewers so much for the time it took to review our manuscript. I welcome any additional questions or comments on our revised manuscript. Thank you.

Round 2

Reviewer 1 Report

I appreciate the time and effort taken by the Author team to revise this manuscript. Likewise, their intention to help develop the evidence base around a rapidly developing topic area is admirable. However, I must note that I still have reservations relating to the description of methods that would allow this paper to be categorized as a systematic review.

Abstract:

  1. b) The study team has revised this section to include a more detailed description of the study methodology.

Insufficient detail still for this section. Not enough to just state PRISMA was used. The authors need to tell the reader what they did briefly e.g. inclusion criteria, study types and dates searched, methods for synthesis…

Additionally, search yield is a result not a method.

25: Typo - “finally” to “final”

  1. c) The study team has revised this section to include a more detailed description of the study results.

Insufficient detail still for this section. Brief summary of characteristics of studies, summary of effect estimates etc. This is a repeat of my previous comment as it has not been addressed.

Methods:

Many of my suggestions/queries about methodology weren't really addressed in this revision and most of those queries still remain. In particular if I look at the PRISMA checklist for reporting there are lots of details lacking that would answer my questions. It is quite time consuming to restate those comments so I will not do so. I would suggest the authors re-consult that checklist and either report what they did accurately in the paper for each item and if they were unable to do something, report this transparently and the reason in the methods. While there is a lack of evidence currently surrounding COVID-19, systematic review methods are standard, their description should not be impacted by this. 

Results:

While authors have stated they accepted editorials and such in the Risk of Bias section of the methods, this needs to be represented in Table 2. The type of evidence is important to the reader's ability to determine and interpret the results of individual studies and how they contribute to the overall outcome of the review. Characteristics of included studies also includes the types of participants/patients that are included in those studies, the components of the intervention etc...this is all important information for the reader to be able to make sense of the results.

It is important to include reasons for exclusion of studies from the results after full-text review. This is also a comment I made last time and inline with the PRISMA checklist.

Would suggest removing the historical articles and keeping the results section purely to discuss the studies returned in the systematic search.

Discussion:

To clarify my previous comment :

"Discussion does not really develop or discuss ideas in relation to the wider scientific literature. It appears to be more of a summary of the main results, in fact something like this would have been beneficial in the results section." The discussion should link the results to the existing literature, even if it is relating the results to other diseases given the lack of publications around COVID.

I wish the authors all the best with their research in this important area.

Author Response

Dear Reviewer 1:

       Our study team is extremely thankful to you for your time and effort. Attached is our revised draft of the manuscript, “The Impact of COVID-19 on Telemedicine Utilization Across Multiple Service Lines in the United States“ for your consideration for publication in Healthcare. We have addressed all of the feedback and recommendations provided by you. In a similar fashion to the previous revised document, the changes are highlighted within the manuscript for your ease in reading. Please see below (in blue text), for a point-by-point response to your comments and concerns. All page numbers refer to the revised manuscript file with tracked changes.

Author response: Thank you so much taking the time to review our article. We hope that this revision improves the overall content of the first draft.

Abstract:

  1. b&c: The study team has completed a near complete rewrite of the abstract. Understandably, due to the 200-word requirement it would be nearly impossible to include all areas the reviewer has requested in the abstract. We have, however, done our best to address each of them in the abstract, while also looking at other examples of abstracts from MDPI Healthcare systematic reviews. We believe our abstract is well in line with other abstracts and meets all necessary requirements.

Methods:

We fully appreciate Reviewer 1’s perspective and recommendations on the Methods section. As such, we have consulted the PRISMA checklist at the behest of Reviewer 1, we have rewritten the entirety of the methods section to match the methods portion of the checklist exactly. We further reviewed the PRISMA publication itself, and many other examples of PRISMA-formatted systematic reviews, and it is our sincere belief that all areas highlighted by Reviewer 1 have been addressed, and that we followed the PRISMA checklist as required. We endeavored to ensure that details were specific and transparent, and that our modifications were fully justifiable and accurate.  

Results:

  • We are grateful to the Reviewer for their valuable recommendations. Thus, we have addressed each of these concerns in the rewritten Methods section. Additionally, the Reviewer is correct when they state that the authors accepted editorials for the systematic review. We have identified which article is in fact, an editorial with the words “(Editorial)” immediately following the article title in Table 2. In this manner the reader may have a better understanding of how this editorial contributes to the review. However, our concern in including such information on this table as the types of participants/ patients and the components of the intervention would result in a number of “Not applicable” for this category as a number of articles had neither of these components inherent in the article. Please recall that this review looked at ALL literature which was available on the subject, not just peer-reviewed articles. We did this to ensure a full inclusion of any possible “tidbits” that might have existed surrounding our chosen topic. We are confident that the reader will be able to make sense of the results in the revised format. We are hopeful the Reviewer agrees with our assessment.
  • At the Reviewer’s recommendation, we have removed the reference to exclusion of studies from the results after full-text review as this more accurately reflects actions taken by the authors during the conduct of the review.
  • At the recommendation of Reviewer 1, we have removed the historical articles in our systematic review, specifically from Table 2, the affinity matrix and the discussion section in order to keep the results section purely to discuss the studies returned in the systematic search. We thank the Reviewer for this valuable recommendation.

Discussion:

At the recommendation of the Reviewer, we have enhanced the discussion section in order to better link the results to the existing literature.

Reviewer 2 Report

Sir,

Authors have significantly improved their paper with a significant effort, that I truly appreciated.

Still (sorry) I've some minor concerns about this study, and more precisely:

1) as previously stated in the initial review, the introduction is a little bit long with some misleading references: as I previously suggested, while it is true that US have spearheaded the implementation of telemedicine, there is a considerable evidence from other continents, such as Australia/Oceania and Europe (lesser islands) that should be at least partially cited across the text and particularly in the introduction and discussion (or rather Authors should more strictly focus on the telemedicine in the US, beginning with the title, e.g. The Impact of COVID-19 on Telemedicine Utilization Across Multiple Service Lines IN THE UNITED STATES)

2) now the selection procedure is more clear and transparent, but Authors should explain in further details what they actually did, now summarized in the sentence: "The final grouping of articles, after using both inclusion and exclusion criteria, were analyzed by all researchers. Multiple consensus meetings were convened in which source material was discussed at length until a definitive pool of literature was deemed worthy of the study". In other words, what did you define as "worthy"? Have you employed a preventive scoring system? Please explain.

3) the abstract is a structured one, please check the authors guidelines if it is appropriate

4) please check the format of tables, that often have retained the original format of Microsoft Word rather than the one recommended by MDPI.

After such improvements, I think that the present paper may deserve a full publication on Healthcare

Author Response

Dear Reviewer 2:

       Our study team is extremely thankful to you for your time and effort. Attached is our revised draft of the manuscript, “The Impact of COVID-19 on Telemedicine Utilization Across Multiple Service Lines in the United States“ for your consideration for publication in Healthcare. We have addressed all of the feedback and recommendations provided by you. In a similar fashion to the previous revised document, the changes are highlighted within the manuscript for your ease in reading. Please see below (in blue text), for a point-by-point response to the reviewers’ comments and concerns. All page numbers refer to the revised manuscript file with tracked changes.

  1. We fully appreciate and accept the Reviewer’s recommendation to highlight the successful telemedicine accomplishments of the international community. We have incorporated and highlighted the very successful telemedicine program found in the islands of Procida and Ischia in the gulf of Naples. At the recommendation of Reviewer 2, we have changed the title of the manuscript to read “The Impact of COVID-19 on Telemedicine Utilization Across Multiple Service Lines IN THE UNITED STATES.”

2) We are grateful to the Reviewer for finding that our revised manuscript’s description of the selection procedure was more clear and transparent. Our revised methods section now better explains in further details what actions we actually executed and addresses the Reviewer’s concerns. Our methodology did not include a preventive scoring system as this was not part of the PRISMA format.

3) The abstract has been reformatted to be in line with the authors guidelines as defined by MDPI.

4) The tables have been reformatted to be in line with the authors guidelines as defined by MDPI.

Again, thank you so much for the time it took to review our manuscript. I welcome any additional questions or comments on our revised manuscript. Thank you.